# The Effect of Ethics in Business on Happiness, Aggressiveness and Inconsistency of Efforts and Rewards

Saif Mahdi Muslim Al-Ameedee and Mahdi Moradi *

Department of Economics and Administrative Sciences, Ferdowsi University of Mashhad,
Mashhad 9177948974, Iran
* Correspondence: mhd_moradi@um.ac.ir

**Abstract:** The present study investigates the effect of business ethics on happiness, aggression and inconsistency of effort and reward of auditors in Iran and Iraq. The statistical population of the present study includes all partners, managers and auditors working in audit institutions in Iran and partners of the audit institutions, assistant auditors, auditors, individual second rank and individual first rank, with a total of 365 questionnaires completed by Iranian respondents out of 450 questionnaires and 250 questionnaires completed by Iraqi respondents out of 350 questionnaires, a total of 615 questionnaires from the two countries in 2022. Also, the methods of variance analysis and ordinary least squares regression and Smart PLS 3 and Stata 15 software were used to analyze the data and test the hypotheses. The results from testing this research's hypotheses indicate a negative and significant relationship between business ethics and aggression, effort-reward mismatch and a positive and significant relationship between business ethics and happiness. Since the current research was conducted in the emerging financial markets of Iran and Iraq, which are highly competitive, along with having special economic conditions, and since the occupation of the ISIS terrorist group, the civil wars in Iraq, severe world economic sanctions against Iran and the global crisis of Covid-19 in Iran and Iraq have led to special conditions, the current research can bring helpful information to readers and help the development of science and knowledge in this field.

**Keywords:** business ethics; happiness; aggressiveness; effort and reward mismatch; auditors

## 1. Introduction

Ethics is a comprehensive subject that covers all aspects of human life, and the ever-increasing growth of human societies and more complex social relationships have created new needs. The emergence of various professions results from efforts to respond to these needs, which are formed with time and changing conditions, and gradually follow the path of transformation and evolution. Due to the necessity of the division of labor and specialization of affairs, these professions are becoming more coherent daily and play an inclusive role in improving the general well-being of societies. The continuation of any profession's life and its members' employment depends on the type and quality of services provided and the reliability and trust gained as a result of providing these services. Reliability and trust are the principal capital of any profession and maintaining them is very important. This requires that the main task and goal of every profession and its members is to serve society, and personal interests are interpreted and pursued only within the framework of providing these services. Auditing as a profession is not separate from this and due to the type and nature of the services it offers, it must have high reliability and trust. Also, the continuation of this reliability and trust and its strengthening depends on the intellectual and practical adherence of the profession's members to its behavioral and ethical standards (Mili et al. 2019).

Independent auditors are part of the corporate governance system of a business entity (Felo 2011) that play an essential role in ensuring that shareholders and other interested

parties outside the company receive transparent information about the company's activities. Also, independent auditors are critical to moving the economy based on trust. Users of the financial information of companies, such as investors, government agencies and the general public, rely on external auditors to provide unbiased audit reports (Kesimli et al. 2018) because this profession is based on professional judgment, and their judgment should be based on reasons and evidence, which means the same evidence collected in audit work from the interaction with the employer's company.

In recent years, one of the essential variables or factors that has attracted the attention of financial and accounting researchers (especially auditors) is ethical decision-making and ethics (Sheehan and Schmidt 2015) and social responsibility in the audit profession regarding compliance with ethics because, according to what was mentioned before (the duty of giving credit to financial statements), the audit profession is in a position where compliance with ethical standards in professional judgments is essential and obvious (Haeridistia and Fadjarenie 2019). Grbac and Lončarić (2009) stated that Croatian managers have a positive attitude toward the importance of ethics and social responsibility for the company's success. Academic research views business ethics and social responsibility as general concepts of doing the right thing, either in terms of positive or negative consequences. Business ethics and social responsibility are independent constructs of business outcomes (Ferrell et al. 2019). These two concepts are closely linked because "a company with a CSR policy must be an ethical company, and an ethical company must be socially responsible" (Fassin et al. 2011). However, corporate social responsibility is defined as a business approach that contributes to sustainable development by providing economic, social and environmental benefits to all stakeholders (Mili et al. 2019), and ethics related to correct behavior is by ethical standards (Mili et al. 2019).

Different decisions, responsibilities and behaviors in an organization have different consequences. Since judgment is one of the main characteristics of the audit profession, and is based on information obtained from companies, it can affect the auditors' judgment (Mansouri et al. 2009). Therefore, it is argued that auditors' view of compliance with ethics in companies may be taken positively or negatively. Forgas (1995) showed that the auditor's negative moods and emotions require more effort and analytical procedures than their positive moods; as a result, more effort in the presence of negative mental states may increase the auditor's performance in providing secondary explanations about fluctuations and distortions in companies. This case will lead to more procedures, higher working hours and, as a result, higher fees, and sometimes increase the level of anxiety perceived by auditors. Also, the auditor performs audit responsibilities through the knowledge created through experience, training and personal characteristics and experiences. Auditors' failure to detect fraud may be influenced by individual and group bias, personal and professional experience, confusion, prejudice, expectations, peer pressure and ability and risk-taking (Kleinman et al. 2012; Kleinman et al. 2020). Besides, the failures of auditors may be derived from the characteristics of the audited company, which can be attributed to non-compliance with ethical principles in business. Compliance with ethics in business is necessary for all professions, including accounting and auditing, because failure to comply with ethics (Barlaup et al. 2009) will expose auditors to a higher litigation risk (Anantharaman et al. 2016), which will result in auditor aggression. Trevino and Brown (2004) stated that paying attention to the issue of unethical behavior in organizations is very important because unethical leadership destroys the organization's reputation, discourages employees and ultimately reduces shareholder value. As a result, auditors need a higher effort to audit such organizations and often face the anxiety caused by auditing such organizations. Hence, around the world, a number of corporate scandals and frauds have inspired many researchers to study the ethical behavior of business leaders and their ethical decision-making (Sharma et al. 2019). These corporate scandals and frauds caused companies to identify their ethical violations and be aware of their social responsibility and ethical behavior (Tu and Lu 2016).

Darcy (2010) emphasized that the work environment of today's organizations is such that people are hesitant to implement true values and ethical standards. Therefore, it is necessary to pay attention to ethics in business in order to provide reasons for improving ethics in organizations because if the organizations follow ethical standards the number of distortions by companies will be reduced. Then, the auditors' attitude towards the reports presented by the companies will be positive. By reviewing the subject literature in the field of ethics, happiness, aggression, effort inconsistency and the reward of auditors, we found that the conducted studies in this field so far mainly include: ethics and depression (Hsin et al. 2016; Rogol 2020); ethical challenges (Marks et al. 2021); ethical issues and the treatment of anxiety (Altis et al. 2014); the impact of professional ethics, independence and auditor experience on audit quality (Haeridistia and Fadjarenie 2019); the ethics and performance of organizations (Goulet 2016); happiness and ethical climate in companies (Kim and Werbach 2016); aggression and morality (Gubler et al. 2018); quality ethics and audit fees (Duong et al. 2022); the effect of ethical behavior and social identity on the performance of auditors. No study has been undertaken to examine the effect of ethics on happiness, aggressiveness and the inconsistency of the efforts and rewards of auditors in two developing countries such as Iran and Iraq with their own political-economic conditions (for example, the occupation of Iraq by the terrorist group ISIS and the civil wars of recent years (Ali Ahmed et al. 2019) and severe economic sanctions imposed on Iran by America and Europe). Therefore, the present research seeks to solve the research gap and contribute to this field's development and body of knowledge by examining such an issue. It can be stated that the mental condition of auditors as members of corporate governance mechanisms is affected by the business ethics of companies. Accordingly, in this research, the answers to two key and basic questions are sought: firstly, whether business ethics affect the psychological conditions of auditors, and secondly whether this influence is different in the two countries of Iran and Iraq or not. In the following sections, the theoretical principles and hypotheses of the study, then the methodology, data analysis and finally the discussion and conclusions of the research findings are discussed.

## 2. Theoretical Principles and Hypothesis Development

Independent auditors play an important role in moving the trust-based economy because users of corporate financial information, such as investors, government agencies and the general public, rely on an external auditor to provide an unbiased audit report (Kesimli et al. 2018). This profession is based on professional judgment. Their judgment should be based on reasons and evidence, which means the same evidence collected in the auditing work from interaction with the employer's company. In recent years, the critical factors that have attracted the attention of financial and accounting researchers (especially auditors) have been ethical decision-making and ethics in the profession (Sheehan and Schmidt 2015) and social responsibility concerning ethics. Since different decisions, responsibilities and behaviors in an organization have different consequences and judgment is one of the main characteristics of the auditing profession, it can affect auditors' judgments (Mansouri et al. 2009). Therefore, it is argued that auditors' views of compliance with ethics in companies may be taken positively or negatively. This profession is one of the professions with the highest stress levels because auditors face conditions of fatigue and depression caused by their work (Public Company Accounting Oversight Board (PCAOB) 2013). After all, the company's environment and conditions influence emotions, moods and psychological conditions during the judgment process. Therefore, identifying the factors affecting the auditor's feelings and mental state seems necessary and should be considered.

Auditors' failures may be derived from the characteristics of the audited company, which can be attributed to non-compliance with ethical principles in business. Compliance with ethics in business is necessary for all professions, including accounting and auditing, because failure to comply with ethics (Barlaup et al. 2009) will expose auditors to a higher risk of litigation (Anantharaman et al. 2016), which will result in auditor aggression. Some models show that effort in the workplace is expended in the context of an organized social

exchange process where adequate rewards in the form of money (salary and benefits), respect (respect and support), or job security and opportunities are expected (Devonish 2018). High effort in combination with low reward is called effort-reward imbalance, and there is empirical support for the view that excessive commitment in a career or business and an imbalance in receiving rewards increases the risk of developing problems such as depression (Hinsch et al. 2019).

Happiness is defined as a combination of feeling good and performing well, which includes experiencing positive emotions such as happiness and satisfaction, developing potential abilities, having control over life, having a positive sense of purpose and experiencing positive relationships (Huppert 2009). Happiness, regardless of how it is created, can improve physical health. The more time a person spends on positive emotions, the less time he leaves for negative emotions. Researchers emphasize that when tension, pain and suffering are reduced, the satisfaction of physical and psychological needs emerges. Heintzelman and Diener (2019) believe that happiness is a type of evaluation that a person makes of his life and includes things such as life satisfaction, excitement and positive mood, lack of depression and anxiety and its various aspects in the form of cognitions and emotions. In the following, the relationship between the variables is discussed.

### 2.1. Explaining the Relationship between Business Ethics and Auditors' Aggressiveness

Based on the recent frauds in financial reports all over the world caused by non-compliance with the regulations and ethical principles and standards of the financial system, some of the causes of the economic crisis, there has been a growing distrust of external auditing. International standards and codes of ethics in business are tools guaranteeing the independence and objectivity of professional accountants when faced with the demands of management (Nerandzic et al. 2012). The importance of ethics in business operations stems from the fact that we face negative tendencies of selfishness, aggression, manipulation, etc., with altruism, tolerance and humanity (Bogdanović 2008).

Business ethics failures can result in huge financial losses for individuals, businesses and society. Companies headquartered in places characterized by more human violence are more likely to fraudulently misrepresent their financial statements and exhibit more aggressive financial reporting. In sum, the exposure to human violence has significant and real effects on individuals' ethical decision-making (Gubler et al. 2018) and, consequently, organizational outcomes. A lack of moral commitment or engaging in unethical behavior not only increases physical aggression towards others, as has been shown, but also facilitates the tendency to lie and cheat (Gubler et al. 2018).

Siegrist (1996) developed the effort-reward imbalance model based on the contrast of external and internal effort with reward, which consists of two leading indicators: the effort-reward ratio (ERR) and overcommitment (a personality trait). According to Siegrist et al. (2004), an imbalance between effort and reward may lead to active distress or aggression by arousing strong negative emotions (Shimazu and de Jonge 2009). The model presented by Siegrist (1996) states that the imbalance between effort and reward is reinforced by overcommitment so that overcommitted employees will respond with stronger reactions to the imbalance of effort and reward compared to less committed employees. (Satoh et al. 2017). Therefore, we expect auditors to experience psychological anxiety or aggression when faced with ethical crises, including companies incorrectly reporting and their lack of accountability for their responsibilities.

Grbac and Lončarić (2009) determined that Croatian managers have a positive attitude towards the importance of ethics and social responsibility for the company's success. Except for age, the relationship between other individual variables and understanding the role of ethics and social responsibility in doing business was not proven. Also, they discovered a positive correlation between the perception of the role of ethics and social responsibility in doing business, profit and work productivity. Gubler et al. (2018) also showed that professional ethics reduce employee conflicts and aggression in organizational environments. Also, the results of Gubler et al. (2018) show that companies located in places

with more human violence are likely to fraudulently distort their financial statements and show more unfair financial reports. The results showed that exposure to human violence has significant and real effects on people's moral decision-making.

Benkraiem et al. (2021) generally state that even if audit standards are strong, they may reduce tax evasion. Ethical corporate behavior has a stronger effect in achieving this goal. In particular, ethical corporate behavior is effective for low- and middle-income countries with low and high levels of investor support and low-performing corporate boards. However, ethical behavior and audit standards are mutually effective for high-income countries and countries with medium-level investor support and boards of directors of medium- and high-performing firms.

Ahmad et al. (2022) found that the aggressiveness of sales managers has an adverse effect on service performance. In research by Akdağcik et al. (2022), they found that as the level of athlete behavior increases, the level of aggression and anger of athletes decreases. Al-Dhubaibi's (2020) findings showed a significant audit expectation gap in Saudi Arabia regarding auditors' responsibilities in general and their responsibility for fraud detection in particular. Contrary to what auditors believe, financial statement users believe auditors should be held accountable for losses to stakeholders if they fail to disclose potential fraud in the audit report or later discover errors in audited financial statements.

According to the stated theoretical principles and background, the first hypothesis of the research is proposed as follows:

**H1.** *Business ethics affect the aggressiveness of auditors.*

*2.2. Explaining the Relationship between Business Ethics and the Mismatch between an Auditor's Effort and Reward*

Auditors' livelihoods and interests rely on receiving fees from their employers (Mautz and Sharaf 1961; Eynon et al. 1997). Auditors as a third party should present their opinion based on ethical reasoning and ethical behavior (Ponemon and Gabhart 1994), which are ethical arguments both by the auditors themselves and imposed by the business environment. Therefore, the auditors' responsibility is influenced by factors (Naslmosavi and Jahanzeb 2017), including ethics in the profession or business. Culpan and Trussel (2005) commented on the bankruptcy of Enron in 2001 and concluded that the bankruptcy resulted from an audit firm's involvement with Enron's leadership to cover up wrongdoing. Enron's auditors made unethical decisions that oppressed Enron's shareholders and other stakeholders (Culpan and Trussel 2005). Unethical audit behavior in the Enron scandal damaged the reputation of many audit firms (Carcello et al. 2005). According to this experience, it can be argued that it is possible that the audit companies were hiding false reports and some immoralities and frauds to receive rewards or fees.

On the one hand, merit-based reward and the competitive promotion system are precious for auditors' business performance; on the other hand, audit firms value their social status, reputation and credibility. In addition, maximizing individual performance indicators, i.e., high audit fees, customer retention and high satisfaction rates, requires accepting a minimum level of risk (Hazgui and Brivot 2020). Therefore, auditors try to reduce the risk of violating ethical norms, especially after the circumstances that arose for companies such as Enron, because it is possible to damage the reputation of companies (Power 2004). Therefore, they increase their efforts to identify such unethical violations in companies, thus protecting their reputation. If the auditor faces pressure from the client and a moral dilemma, he will be greatly influenced by the auditor's reasoning and moral belief (Naslmosavi and Jahanzeb 2017). The difference between the levels of accountability provided by the auditor as anticipated and expected by independent accountants and users of financial statements creates an expectation gap (Haniffa and Hudaib 2007). The auditor increases his effort according to the amount of this expectation gap, the existing risk and the resources spent for the audit. According to the effort spent, he expects to receive a higher fee and reward, but the reward may be proportional to the effort spent.

Using comparative data from five countries, Siegrist et al. (2004) examined the psychometric properties of the effort-reward imbalance (ERI) model at work. In this model, chronic work-related stress is identified as the reason for the imbalance between the high efforts expended and low rewards received.

Tanimoto et al. (2023) found that reward was negatively related to burnout and health and showed that the interaction between effort and reward was significantly related to all outcomes among employees with permanent contracts. Still, it was not meaningful for those who had a fixed-term contract. Devonish's (2018) research showed that the partial mediation model is superior to the full mediation model, indicating that job satisfaction only partially mediates the relationships between effort-reward imbalance and burnout, intention to leave and mental health.

According to the stated theoretical principles and background, the second hypothesis of the research is proposed as follows:

**H2.** *Business ethics affect the effort inconsistency and reward of auditors.*

*2.3. Explaining the Relationship between Business Ethics and Happiness*

Compliance with business ethics in the professional environment can lead to happiness among the members of the profession (De Neve and Ward 2017). Maslow (1943) proposes a hierarchy of needs that people must achieve and suggests that when needs above a particular hierarchy level are met, they feel happy. Compliance with ethics in any profession creates a sense of satisfaction and gratitude among employees (De Neve and Ward 2017). In the present era, the primary concern of managers is the emergence of issues and problems that arise due to the non-observance of ethical principles by the employees of units. Therefore, to carry out their affairs and work, in addition to paying attention to their ethical code and the rules defined for their organization, the units and organizations also pay attention to ethical issues to help them achieve their goals because ethical principles are one of the most critical indicators that help in creating a suitable and calm environment and by themselves lead to an increase in group interactions among the employees of organizations and, in this way, also lead to an increase in their efficiency and productivity (Gubler et al. 2018).

As with other organizations, the proper performance of independent auditors and increasing audit quality have become necessary in audit firms. Because the primary duty of auditors is to give credit to the forms that users rely on in making decisions, also auditors are the representatives of shareholders in organizations, supporting their interests in front of managers, so in the current situation failure to comply with ethical indicators has become one of the most critical concerns of governmental and non-governmental institutions. One of the ways to reduce these concerns is for audit institutions to comply with ethical behaviors as best as possible (Duong et al. 2022). The supervisor, partner, or superior in the audit firm behaves according to the code of professional conduct and ethical standards with all subordinates, and they are also affected by their behavior from several dimensions: firstly, subordinates feel security and justice; secondly, they know their manager, supervisor or superior as a person who is committed to ethical affairs, so they cooperate with him in order to achieve the goals of the organization (Duong et al. 2022). This is because the heads of groups (leaders and managers) are responsible for their subordinates, so they need ethics and methods more than any other person. Because these people are in a position that is different from others and from where they can be the origin of many changes, leaders can convey the importance of ethics to their subordinates with ethical behavior, trust and loyalty (Tutar et al. 2011). Elçi and Alpkan (2009) showed that when employees realize that their supervisor behaves fairly, honestly and trustfully, they gain a positive attitude towards work and the organization, so their performance is higher.

In other words, a socially responsible company (with a corporate social responsibility policy) must be ethical, and an ethical company must be socially responsible (Fassin et al. 2011). Corporate social responsibility (CSR) is a business approach that contributes to

sustainable development by providing all stakeholders with economic, social and environmental benefits. It is about accountability to all stakeholders, not just shareholders (Mili et al. 2019). Friedman (1970) suggests that CSR practices divert the firm from its goals of profit generation and shareholder wealth maximization. However, adopting ethical principles may affect its image in the business world and hence can be seen as a significant factor in attracting new partners and investors. Based on these claims, it can be argued that ethical principles can positively affect companies' financial performance (Mili et al. 2019).

In-depth study and insight into ethical behavior are desirable because they affect employee attitudes and behavioral outcomes such as organizational commitment, deviance, job satisfaction, employee happiness, role performance and creativity. For example, managers aware of internal information related to the company's performance that is not available to investors or the general public may be tempted to use this private information for their own personal gain, which may lead to violations of ethical norms. (Micewski and Troy 2007). All of these also affect the auditor's view of the audited organization in such a way that a positive view of the positive results of the company in case of compliance with ethics will cause the auditor to be satisfied with the audit, as a result of which the auditor's credibility will increase, and the auditor will be happy. On the contrary, negative insight regarding the company's results, in the case of non-compliance with ethics, causes the auditor's dissatisfaction with the audit. As a result, the auditor's credit will be reduced, which will cause the auditor's dissatisfaction.

Shafer et al. (2001), by surveying a random sample of members of the American Institute of Certified Public Accountants (AICPA), assessed their value preferences and reactions to an ethical dilemma involving client pressure for aggressive financial reporting. Contrary to the hypotheses of their study, personal value preferences did not affect the auditors' perceptions of the severity of the ethical dilemma. Also, as hypothesized, moral severity perceptions influenced moral judgments and behavioral intentions.

Jeong et al. (2022) showed that promoting social responsibility can positively affect team members' happiness at work and in the organization. The findings of the Ferrell et al. (2019) study provide new insights into customer expectations and perceptions of corporate social responsibility and business ethics behavior and they conclude that although social responsibility attitudes are necessary, customers perceive business ethics as a critical behavior of their perceptions toward brand value. The observance of audit institutions' ethics can cause employees a sense of satisfaction and increase their happiness (Duska and Duska and Duska 2003). Ferrell et al. (2019) showed that business ethics significantly impacted customer behavior and perception.

According to the stated theoretical principles and background, the third hypothesis of the research is proposed as follows:

**H3.** *Business ethics affect the happiness of auditors.*

### 3. Research Methodology

In order to collect data and information for analysis, a questionnaire with a five-point Likert scale (1 to 5) from very little to very much was used. Also, the methods of variance analysis and ordinary least squares regression and Smart PLS 3 and Stata 15 software were used to analyze the data and test the hypotheses. The dependent variable of this research is the auditors' psychological conditions, including happiness, aggressiveness and inconsistency between effort and reward. To measure happiness (WEL), the Hills and Argyle (2002) questionnaire was used, which includes 8 questions, to measure aggression (AGR), the Pettersen et al. (2018) questionnaire was used, which contains 29 questions and to measure inconsistency between efforts and reward (MER) the Li et al. (2017) questionnaire was used, which includes 16 questions. The independent variable of this research is business ethics (BET), measured by the Canizales business ethics questionnaire (Canizales 2009), including 16 questions. The Smart PLS method was also used for data

analysis because of the sample size, the appropriate predictive power of this method and its exploratory analysis.

*Statistical Population*

The current questionnaire was distributed among the respondents on December 27 2021, so it can be said that the current research was conducted from the beginning of 2022. The information and answers to the questionnaires were collected until the end of March 2022. The statistical population of the present study includes all partners, managers and auditors working in audit institutions in Iran and partners of the audit institution, assistant auditors, auditors, individual second rank and individual first rank for Iraq, with a total of 365 questionnaires by Iranian respondents out of 450 questionnaires and 250 questionnaires completed by Iraqi respondents out of 350 questionnaires, a total of 615 questionnaires from the two countries in 2022.

Table 1 presents the characteristics of questionnaire respondents in the two countries of Iran and Iraq.

**Table 1.** The number and education of Iranian questionnaire respondents.

| Job Position | No. | High Diploma | MSC | Masters | PhD. |
|---|---|---|---|---|---|
| Partner of the audit institutions | 41 | | 3 | 8 | 30 |
| Manager of the audit institutions | 38 | | 1 | 12 | 25 |
| Senior supervisor | 67 | | 4 | 54 | 9 |
| Supervisor | 78 | | 38 | 34 | 6 |
| Senior auditor | 36 | | 8 | 21 | 7 |
| Auditor | 49 | | 19 | 28 | 2 |
| Auditor's assistant | 56 | | 51 | 5 | 0 |
| Total | 365 | | 124 | 162 | 79 |
| The number and education of Iraqi questionnaire respondents | | | | | |
| Partner of the audit institutions | 16 | 4 | 0 | 8 | 4 |
| Assistant auditor | 4 | 0 | 3 | 1 | 0 |
| Auditor | 107 | 1 | 23 | 72 | 11 |
| Individual second rank | 48 | 0 | 0 | 28 | 20 |
| Individual first rank | 75 | 17 | 0 | 18 | 40 |
| Total | 250 | 22 | 26 | 127 | 75 |

## 4. Research Findings

### 4.1. Descriptive Statistics

A total of 450 questionnaires were distributed in Iran and 350 questionnaires in Iraq, of which 365 questionnaires were completed in Iran and 250 questionnaires in Iraq. The questionnaire completion ratio was 0.81% in Iran and 0.71% in Iraq. These questionnaires were filled out by all the partners, managers and auditors working in auditing firms in Iran and the partners of the auditing firm, assistant auditors, individual second rank and individual first rank for Iraq working in companies active in the stock exchanges of Iran and Iraq.

As shown in Table 2, in terms of gender the number of male respondents to the questionnaire in Iran is more than female respondents; most are between 30 and 39 years old. Most of the group has a bachelor's degree, and they work in the accounting field with experience of between 1 and 5 years. Also, most of the respondents to the questionnaire in Iraq were men, and most of the respondents were between 40 and 49 years old. Most of the group have postgraduate education and work in accounting with 11 to 15 years of experience.

**Table 2.** The frequency demographic data of Iran and Iraq.

|  | No. | Percentage | No. | Percentage |
|---|---|---|---|---|
| | | Gender: | | |
| Male | 307 | 84.11 | 186 | 74.40 |
| Female | 58 | 15.89 | 64 | 25.60 |
| Total | 365 | 100 | 250 | 100 |
| | | Age: | | |
| 20 to 29 years | 130 | 35.69 | 17 | 6.80 |
| 30 to 39 years | 158 | 43.29 | 96 | 38.40 |
| 40 to 49 years | 60 | 16.44 | 102 | 40.80 |
| Above 50 years | 17 | 4.66 | 35 | 14 |
| Total | 365 | 100 | 250 | 100 |
| | | Work Experience: | | |
| 1 to 5 years | 143 | 39.18 | 9 | 3.60 |
| 6 to 10 years | 110 | 30.14 | 56 | 22.40 |
| 11 to 15 years | 42 | 11.51 | 78 | 31.20 |
| 16 to 20 years | 30 | 8.22 | 57 | 22.80 |
| 20 to 25 years | 20 | 5.48 | 23 | 9.20 |
| 26 to 30 years | 20 | 5.48 | 27 | 10.80 |
| Total | 365 | 100 | 250 | 100 |

Table 3 shows the descriptive statistics of business ethics data in Iran and Iraq. According to the table, the average and median of the answers given by all the partners, managers and supervisors of audit institutions in Iran are 4, which is a high choice, and the mode in the received answers is also an increased choice. After that, the frequency, including the number and percentage of each of the responses to the Iranian questionnaire, which 365 people completed, was stated in order. In Iraq, mode and median are often chosen as 4 or 5, which are many and very many options. Therefore, we can point out the importance and role of business ethics in Iraq.

**Table 3.** The Descriptive statistics of data related to variables in Iran and Iraq.

| The Descriptive Statistics of Data Related to Business Ethics in Iran and Iraq | | | | | | | | | |
|---|---|---|---|---|---|---|---|---|---|
| Question | Mean | Median | Mode | Std. dev. | Question | Mean | Median | Mode | Std. dev. |
| | | | | Iran | | | | | |
| Q1 | 4.180 | 4 | 4 | 0.730 | Q9 | 3.770 | 4 | 5 | 1.230 |
| Q2 | 4.060 | 4 | 4 | 0.770 | Q10 | 3.790 | 4 | 5 | 1.200 |
| Q3 | 4.240 | 4 | 4 | 0.620 | Q11 | 3.620 | 4 | 4 | 1.130 |
| Q4 | 4.300 | 4 | 4 | 0.640 | Q12 | 3.730 | 4 | 4 | 1.170 |
| Q5 | 3.880 | 4 | 4 | 0.890 | Q13 | 3.820 | 4 | 4 | 1.140 |
| Q6 | 2.200 | 2 | 1 | 1.230 | Q14 | 3.700 | 4 | 4 | 1.100 |
| Q7 | 3.740 | 4 | 4 | 0.910 | Q15 | 3.450 | 3 | 3 | 1.010 |
| Q8 | 4.180 | 4 | 4 | 0.600 | Q16 | 3.490 | 3 | 3 | 0.950 |
| | | | | Iraq | | | | | |
| Q1 | 3.620 | 5 | 5 | 1.470 | Q9 | 3.740 | 4 | 4 | 1.120 |
| Q2 | 4.160 | 4 | 4 | 0.620 | Q10 | 4.180 | 4 | 4 | 0.800 |
| Q3 | 4.490 | 5 | 5 | 0.610 | Q11 | 4.040 | 4 | 4 | 0.830 |
| Q4 | 4.410 | 4 | 5 | 0.650 | Q12 | 4.400 | 4 | 5 | 0.660 |
| Q5 | 4.200 | 4 | 4 | 0.810 | Q13 | 4.400 | 5 | 5 | 0.690 |
| Q6 | 2.860 | 3 | 3 | 1.130 | Q14 | 4.160 | 4 | 4 | 0.750 |
| Q7 | 4.140 | 4 | 4 | 0.670 | Q15 | 4.240 | 4 | 4 | 0.800 |
| Q8 | 4.480 | 5 | 5 | 0.600 | Q16 | 3.870 | 4 | 4 | 0.900 |

**Table 3.** *Cont.*

| The Descriptive Statistics of Data Related to Business Ethics in Iran and Iraq | | | | | | | | | |
|---|---|---|---|---|---|---|---|---|---|
| **Question** | **Mean** | **Median** | **Mode** | **Std. dev.** | **Question** | **Mean** | **Median** | **Mode** | **Std. dev.** |
| The Descriptive statistics of data related to aggression in Iran and Iraq | | | | | | | | | |
| Question | Mean | Median | Mode | Std. dev. | Question | Mean | Median | Mode | Std. dev. |
| Iran | | | | | | | | | |
| Q1 | 2.500 | 2 | 2 | 1.080 | Q16 | 2.580 | 2 | 2 | 1.300 |
| Q2 | 2.220 | 2 | 2 | 1.280 | Q17 | 2.410 | 2 | 1 | 1.370 |
| Q3 | 2.290 | 2 | 1 | 1.250 | Q18 | 3.130 | 3 | 4 | 1.290 |
| Q4 | 2.210 | 2 | 2 | 1.260 | Q19 | 2.540 | 2 | 2 | 1.250 |
| Q5 | 2.400 | 2 | 2 | 1.250 | Q20 | 2.370 | 2 | 2 | 1.270 |
| Q6 | 2.220 | 2 | 2 | 1.300 | Q21 | 2.240 | 2 | 1 | 1.260 |
| Q7 | 3.360 | 4 | 4 | 1.320 | Q22 | 2.280 | 2 | 2 | 1.280 |
| Q8 | 2.100 | 2 | 2 | 1.180 | Q23 | 2.690 | 2 | 2 | 1.290 |
| Q9 | 2.010 | 2 | 1 | 1.220 | Q24 | 2.540 | 2 | 2 | 1.180 |
| Q10 | 3.390 | 4 | 4 | 1.190 | Q25 | 2.450 | 2 | 2 | 1.240 |
| Q11 | 2.840 | 3 | 2 | 1.230 | Q26 | 2.540 | 2 | 2 | 1.320 |
| Q12 | 3.130 | 3 | 4 | 1.120 | Q27 | 2.400 | 2 | 2 | 1.210 |
| Q13 | 2.870 | 3 | 4 | 1.410 | Q28 | 2.250 | 2 | 2 | 1.250 |
| Q14 | 3.810 | 4 | 4 | 0.990 | Q29 | 2.600 | 2 | 2 | 1.280 |
| Q15 | 2.320 | 2 | 1 | 1.260 | | | | | |
| Iraq | | | | | | | | | |
| Q1 | 2.510 | 2 | 3 | 1.030 | Q16 | 3.730 | 4 | 4 | 1.090 |
| Q2 | 1.730 | 1 | 1 | 1.000 | Q17 | 2.600 | 3 | 1 | 1.300 |
| Q3 | 2.590 | 3 | 1 | 1.320 | Q18 | 2.860 | 3 | 3 | 1.180 |
| Q4 | 1.940 | 2 | 1 | 1.100 | Q19 | 2.000 | 2 | 1 | 1.050 |
| Q5 | 2.300 | 2 | 1 | 1.190 | Q20 | 1.950 | 1 | 1 | 1.160 |
| Q6 | 1.700 | 1 | 1 | 1.040 | Q21 | 2.100 | 2 | 1 | 1.190 |
| Q7 | 3.570 | 4 | 4 | 1.350 | Q22 | 1.760 | 1 | 1 | 1.000 |
| Q8 | 1.430 | 1 | 1 | 0.860 | Q23 | 2.750 | 3 | 3 | 1.170 |
| Q9 | 1.730 | 1 | 1 | 1.020 | Q24 | 2.440 | 2 | 3 | 1.130 |
| Q10 | 3.360 | 3 | 3 | 1.040 | Q25 | 2.490 | 3 | 3 | 1.180 |
| Q11 | 2.910 | 3 | 3 | 0.980 | Q26 | 2.460 | 2 | 2 | 1.250 |
| Q12 | 3.100 | 3 | 3 | 1.020 | Q27 | 2.610 | 3 | 2 | 1.140 |
| Q13 | 2.910 | 3 | 3 | 1.040 | Q28 | 1.850 | 2 | 1 | 1.070 |
| Q14 | 3.850 | 4 | 4 | 0.800 | Q29 | 2.800 | 3 | 3 | 1.150 |
| Q15 | 3.410 | 4 | 4 | 1.220 | | | | | |
| The Descriptive statistics of the data related to the mismatch between effort and reward in Iran and Iraq | | | | | | | | | |
| Question | Mean | Median | Mode | Std. dev. | Question | Mean | Median | Mode | Std. dev. |
| Iran | | | | | | | | | |
| Q1 | 3.020 | 3 | 2 | 1.220 | Q9 | 3.520 | 4 | 4 | 1.090 |
| Q2 | 3.010 | 3 | 2 | 1.190 | Q10 | 3.280 | 3 | 4 | 1.200 |
| Q3 | 3.400 | 4 | 4 | 1.020 | Q11 | 2.850 | 3 | 2 | 1.050 |
| Q4 | 3.610 | 4 | 4 | 0.960 | Q12 | 3.170 | 3 | 4 | 1.120 |
| Q5 | 2.950 | 3 | 2 | 1.140 | Q13 | 3.050 | 3 | 4 | 1.180 |
| Q6 | 2.930 | 3 | 2 | 1.140 | Q14 | 3.220 | 3 | 4 | 1.150 |
| Q7 | 2.570 | 2 | 2 | 1.230 | Q15 | 3.050 | 3 | 4 | 1.080 |
| Q8 | 3.710 | 4 | 4 | 0.990 | Q16 | 3.250 | 3 | 4 | 1.090 |

**Table 3.** *Cont.*

| The Descriptive Statistics of Data Related to Business Ethics in Iran and Iraq | | | | | | | | | |
|---|---|---|---|---|---|---|---|---|---|
| Question | Mean | Median | Mode | Std. dev. | Question | Mean | Median | Mode | Std. dev. |
| Iraq | | | | | | | | | |
| Q1 | 3.630 | 4 | 4 | 0.970 | Q9 | 2.850 | 3 | 3 | 1.060 |
| Q2 | 3.390 | 3 | 4 | 1.000 | Q10 | 3.610 | 4 | 4 | 1.070 |
| Q3 | 4.020 | 4 | 4 | 0.840 | Q11 | 3.440 | 3 | 4 | 0.930 |
| Q4 | 4.120 | 4 | 4 | 0.730 | Q12 | 3.820 | 4 | 4 | 0.850 |
| Q5 | 3.350 | 3 | 3 | 1.050 | Q13 | 2.110 | 2 | 1 | 1.090 |
| Q6 | 3.490 | 4 | 4 | 1.000 | Q14 | 3.680 | 4 | 4 | 0.830 |
| Q7 | 3.240 | 3 | 3 | 1.070 | Q15 | 3.260 | 3 | 3 | 1.040 |
| Q8 | 3.990 | 4 | 4 | 0.830 | Q16 | 3.590 | 4 | 4 | 1.000 |
| The Descriptive statistics of the data related to the happiness of Iran and Iraq | | | | | | | | | |
| Question | Mean | Median | Mode | Std. dev. | Question | Mean | Median | Mode | Std. dev. |
| Iran | | | | | | | | | |
| Q1 | 2.600 | 2 | 2 | 1.190 | Q5 | 3.720 | 4 | 4 | 0.980 |
| Q2 | 3.570 | 4 | 4 | 0.940 | Q6 | 3.780 | 4 | 4 | 0.990 |
| Q3 | 3.480 | 4 | 4 | 1.000 | Q7 | 3.830 | 4 | 4 | 0.880 |
| Q4 | 2.570 | 2 | 2 | 1.120 | Q8 | 2.580 | 2 | 2 | 1.090 |
| Iraq | | | | | | | | | |
| Q1 | 2.640 | 3 | 3 | 1.200 | Q5 | 3.840 | 4 | 4 | 0.700 |
| Q2 | 3.930 | 4 | 4 | 0.800 | Q6 | 3.860 | 4 | 4 | 0.720 |
| Q3 | 3.700 | 4 | 4 | 0.870 | Q7 | 3.940 | 4 | 4 | 0.710 |
| Q4 | 3.460 | 3 | 3 | 0.880 | Q8 | 2.890 | 3 | 3 | 1.040 |

Table 3 shows the descriptive statistics of aggression data in Iran and Iraq. The mean and median of the responses provided by all the partners, managers and supervisors of audit institutions in Iran are 2, which is low, and the mode in the received responses is also low. The mean and median of the answers provided by all the partners, managers and supervisors of audit institutions in Iraq are 2, representing a low option, and the mode in the received answers is also very low.

Table 3 shows the descriptive statistics of effort and reward mismatch in Iran and Iraq. The mean and median of the responses provided by all the partners, managers and supervisors of audit institutions in Iran are 3, which is the medium option, and the mode in the received responses is also the high option. The average and median of the responses provided by all the partners, managers and supervisors of audit institutions in Iraq are 4, which is a large number of choices. The mode in the received responses also has a large number of choices.

Table 3 shows the descriptive statistics of the happiness data of the two countries of Iran and Iraq. The mean and median of the responses provided by all the partners, managers and supervisors of audit institutions in Iran are 4, which is a large number of choices. The mode in the received responses also has a large number of choices. Also, the mean and median of the responses provided by all the partners, managers and supervisors of audit institutions in Iraq are 4, which is a high choice, and the mode in the received answers is also high.

*4.2. Inferencing Data*

Before examining the research hypotheses, the validity and reliability of the research questionnaire are examined. When enough evidence has been accumulated about the validity and reliability of the questionnaires, the research hypotheses can be evaluated correctly. The results of these cases are shown in Table 4 as indices of the Cronbach's alpha coefficient, composite reliability coefficient and average variance extracted. The alpha coefficient ranges from 0 to 1. The alpha coefficient for Iran's questionnaire is equal to 0.835,

and for Iraq's questionnaire is equal to 0.811, which is in the appropriate range. To evaluate the validity of the construct, the average variance index was extracted, and the Fornell and Larcker criteria were used. The AVE index in Table 4 states that the average extracted variance of each model dimension has a value greater than 0.5; therefore, the convergent validity of the model is confirmed. According to Table 4, the AVE value for the model's variables is higher than 0.5, so it can be said that the cross-validation-communality was used in the convergence validity of the measurement model.

**Table 4.** The Reliability and validity findings of the research.

| Questionnaire | Cronbach's Alpha | Composite Reliability Coefficient | AVE |
|---|---|---|---|
| Iran | 0.835 | 0.854 | 0.690 |
| Iraq | 0.811 | 0.847 | 0.642 |

In order to measure the goodness of fit of the current research, two indices were used, the results of which are presented in Table 5. It can be concluded that the model fit is suitable for the data of Iran and Iraq, and the accumulated data and the obtained results are reliable.

**Table 5.** The goodness of fit criteria.

| Criteria | Value | | | Acceptable Value |
|---|---|---|---|---|
| | Iran | Iraq | Accumulated | |
| Chi-square index | 0.040 | 0.046 | 0.042 | Less than 0.08 |
| NFI index | 0.902 | 0.905 | 0.904 | The closer to one, the better |
| Index Q2 | More than 0.50 for all variables | | | More than 0.30 |

In Table 6, the results of the equality test of the mean of Iran and Iraq questionnaires are displayed in order to investigate the impact of business ethics on the psychological conditions of auditors.

**Table 6.** The results of variance analysis of the mean of business ethics questions in Iran and Iraq.

| Question | Iran Mean | Iraq Mean | $\mu_{Iran} > \mu_{Iraq}$ | $\mu_{Iran} \neq \mu_{Iraq}$ | $\mu_{Iran} < \mu_{Iraq}$ |
|---|---|---|---|---|---|
| Q1 | 4.18 | 3.62 | 0.000 | 0.000 | 1.000 |
| Q2 | 4.06 | 4.16 | 0.636 | 0.729 | 0.365 |
| Q3 | 4.24 | 4.49 | 1.000 | 0.000 | 0.000 |
| Q4 | 4.30 | 4.41 | 0.938 | 0.124 | 0.062 |
| Q5 | 3.88 | 4.20 | 1.000 | 0.000 | 0.000 |
| Q6 | 2.20 | 2.86 | 1.000 | 0.000 | 0.000 |
| Q7 | 3.74 | 4.14 | 1.000 | 0.000 | 0.000 |
| Q8 | 4.18 | 4.48 | 1.000 | 0.000 | 0.000 |
| Q9 | 3.77 | 3.74 | 0.614 | 0.772 | 0.386 |
| Q10 | 3.79 | 4.18 | 1.000 | 0.000 | 0.000 |
| Q11 | 3.62 | 4.04 | 0.999 | 0.000 | 0.000 |
| Q12 | 3.73 | 4.40 | 1.000 | 0.000 | 0.000 |
| Q13 | 3.82 | 4.40 | 1.000 | 0.000 | 0.000 |
| Q14 | 3.70 | 4.16 | 1.000 | 0.000 | 0.000 |
| Q15 | 3.45 | 4.24 | 1.000 | 0.000 | 0.000 |
| Q16 | 3.49 | 3.87 | 1.000 | 0.000 | 0.000 |

**Table 6.** *Cont.*

| Question | Iran Mean | Iraq Mean | $\mu_{Iran} > \mu_{Iraq}$ | $\mu_{Iran} \neq \mu_{Iraq}$ | $\mu_{Iran} < \mu_{Iraq}$ |
|---|---|---|---|---|---|
| The results of variance analysis of the mean of the aggression questions of Iran and Iraq | | | | | |
| Question | Iran mean | Iraq mean | $\mu_{Iran} > \mu_{Iraq}$ | $\mu_{Iran} \neq \mu_{Iraq}$ | $\mu_{Iran} < \mu_{Iraq}$ |
| Q1 | 2.500 | 2.510 | 0.224 | 0.457 | 0.254 |
| Q2 | 2.220 | 1.730 | 0.000 | 0.000 | 1.000 |
| Q3 | 2.290 | 2.590 | 0.157 | 0.258 | 0.458 |
| Q4 | 2.210 | 1.940 | 0.000 | 0.000 | 1.000 |
| Q5 | 2.400 | 2.300 | 0.158 | 0.448 | 0.252 |
| Q6 | 2.220 | 1.700 | 0.000 | 0.000 | 1.000 |
| Q7 | 3.360 | 3.570 | 1.000 | 0.000 | 0.000 |
| Q8 | 2.100 | 1.430 | 0.000 | 0.000 | 1.000 |
| Q9 | 2.010 | 1.730 | 0.000 | 0.000 | 1.000 |
| Q10 | 3.390 | 3.360 | 0.558 | 0.745 | 0.584 |
| Q11 | 2.840 | 2.910 | 0.475 | 0.485 | 0.406 |
| Q12 | 3.130 | 3.100 | 0.875 | 0.784 | 0.657 |
| Q13 | 2.870 | 2.910 | 0.228 | 0.205 | 0.246 |
| Q14 | 3.810 | 3.850 | 0.338 | 0.486 | 0.518 |
| Q15 | 2.320 | 3.410 | 1.000 | 0.000 | 0.000 |
| Q16 | 2.580 | 3.730 | 0.000 | 0.000 | 1.000 |
| Q17 | 2.410 | 2.600 | 0.105 | 0.119 | 0.128 |
| Q18 | 3.130 | 2.860 | 0.000 | 0.000 | 1.000 |
| Q19 | 2.540 | 2.000 | 0.000 | 0.000 | 1.000 |
| Q20 | 2.370 | 1.950 | 0.000 | 0.000 | 1.000 |
| Q21 | 2.240 | 2.100 | 0.187 | 0.149 | 0.166 |
| Q22 | 2.280 | 1.760 | 0.000 | 0.000 | 1.000 |
| Q23 | 2.690 | 2.750 | 0.227 | 0.257 | 0.248 |
| Q24 | 2.540 | 2.440 | 0.364 | 0.335 | 0.398 |
| Q25 | 2.450 | 2.490 | 0.874 | 0.805 | 0.849 |
| Q26 | 2.540 | 2.460 | 0.258 | 0.246 | 0.289 |
| Q27 | 2.400 | 2.610 | 0.002 | 0.026 | 1.000 |
| Q28 | 2.250 | 1.850 | 0.000 | 0.000 | 1.000 |
| Q29 | 2.600 | 2.800 | 0.158 | 0.109 | 0.118 |
| The results of the variance analysis test of the mean of the mismatch between efforts and rewards in Iran and Iraq | | | | | |
| Question | Iran mean | Iraq mean | $\mu_{Iran} > \mu_{Iraq}$ | $\mu_{Iran} \neq \mu_{Iraq}$ | $\mu_{Iran} < \mu_{Iraq}$ |
| Q1 | 3.020 | 3.630 | 1.000 | 0.000 | 0.000 |
| Q2 | 3.010 | 3.390 | 1.000 | 0.000 | 0.000 |
| Q3 | 3.400 | 4.020 | 1.000 | 0.000 | 0.000 |
| Q4 | 3.610 | 4.120 | 1.000 | 0.000 | 0.000 |
| Q5 | 2.950 | 3.350 | 1.000 | 0.000 | 0.000 |
| Q6 | 2.930 | 3.490 | 1.000 | 0.000 | 0.000 |
| Q7 | 2.570 | 3.240 | 1.000 | 0.000 | 0.000 |
| Q8 | 3.710 | 3.990 | 0.548 | 0.335 | 0.189 |
| Q9 | 3.520 | 2.850 | 0.000 | 0.000 | 1.000 |
| Q10 | 3.280 | 3.610 | 1.000 | 0.000 | 0.000 |
| Q11 | 2.850 | 3.440 | 1.000 | 0.000 | 0.000 |
| Q12 | 3.170 | 3.820 | 1.000 | 0.000 | 0.000 |
| Q13 | 3.050 | 2.110 | 0.000 | 0.000 | 1.000 |
| Q14 | 3.220 | 3.680 | 1.000 | 0.000 | 0.000 |
| Q15 | 3.050 | 3.260 | 0.148 | 0.108 | 0.117 |
| Q16 | 3.250 | 3.590 | 1.000 | 0.000 | 0.000 |

**Table 6.** *Cont.*

| Question | Iran Mean | Iraq Mean | $\mu_{Iran} > \mu_{Iraq}$ | $\mu_{Iran} \neq \mu_{Iraq}$ | $\mu_{Iran} < \mu_{Iraq}$ |
|---|---|---|---|---|---|
| The results of the analysis of variance of the mean of happiness questions of Iran and Iraq | | | | | |
| Question | Iran mean | Iraq mean | $\mu_{Iran} > \mu_{Iraq}$ | $\mu_{Iran} \neq \mu_{Iraq}$ | $\mu_{Iran} < \mu_{Iraq}$ |
| Q1 | 2.600 | 2.640 | 0.148 | 0.153 | 0.192 |
| Q2 | 3.570 | 3.930 | 1.000 | 0.000 | 0.000 |
| Q3 | 3.480 | 3.700 | 1.000 | 0.000 | 0.000 |
| Q4 | 2.570 | 3.460 | 1.000 | 0.000 | 0.000 |
| Q5 | 3.720 | 3.840 | 0.447 | 0.475 | 0.408 |
| Q6 | 3.780 | 3.860 | 0.358 | 0.306 | 0.349 |
| Q7 | 3.830 | 3.940 | 0.249 | 0.241 | 0.208 |
| Q8 | 2.580 | 2.890 | 0.106 | 0.118 | 0.107 |

Table 6 compares Iran and Iraq's average responses received on business ethics. In comparing the averages, the opposite hypothesis is considered in three ways of inequality, Iran's largeness of average received responses and Iraq's largeness of average received responses. In most of the questions, the average answers received by Iraq were higher than Iran. Therefore, business ethics in Iraq has a more prominent role than in Iran. On the contrary, regarding the second question, the average answers received by the two countries are the same. Therefore, the essential and the effective role in the work environment is the same from the point of view of the respondents of Iraq and Iran. Also, in terms of the question, 'To what extent are you looking for a higher rank and position in your current role?' the same answer was given by both groups. On the contrary, for the first question the average answers received in Iran were higher than in Iraq. Therefore, compared to Iraqi respondents, Iranian respondents perform their work duties with more caution and care.

Table 6 compares the average responses to aggression questions for Iran and Iraq. This section contains 29 questions. Regarding half of the questions in this section, the average of the two groups is equal. For questions 7 and 15, the average aggression of Iraq is lower than that of Iran. For 12 questions, Iran's average was lower than Iraq's.

Table 6 includes the average answers received to the questions about a mismatch between the efforts and rewards of the two countries. This section consists of 16 questions. The average of these two countries is also significantly different from each other. The average mismatch between effort and reward is lower in Iran than in Iraq. Only for two questions, 8 and 15, is the average of the two countries is the same. In addition, regarding questions 9 and 13, Iraq's average was lower than Iran's.

Table 6 includes the average answers to the happiness questions of the two countries. This section consists of eight questions. The average of these two countries is different from each other on three out of the eight questions, and they are equal to each other on the other five questions. Regarding questions 2, 3 and 4, the average of Iran regarding happiness is lower than Iraq's.

Psychological conditions include happiness, aggression and inconsistency between effort and reward. Each of these factors has been obtained through several questions through averaging. Business ethics itself is obtained through 16 questions. The component of happiness was obtained through 8 questions, aggression through 29 questions and inconsistency between effort and reward through 16 questions. Table 7 shows the components of each part of the questionnaire and the number of questions that make them up.

In addition, in Table 7, Cronbach's alpha of each part of the questionnaire is also calculated. Considering that Cronbach's alpha was calculated between 0.839 and 0.892 for Iran's questionnaire and between 0.812 and 0.884 for Iraq, the questionnaires of both countries have a suitable internal structure.

**Table 7.** The components, the number of questions, Cronbach's alpha and factor analysis results.

| Variable | Components | Questions | Iran | | Iraq | |
|---|---|---|---|---|---|---|
| | | | Cronbach's Alpha | Factor Analysis | Cronbach's Alpha | Factor Analysis |
| Psychological conditions | Business ethics | 16 | 0.839 | 0.792–0887 | 0.884 | 0.799–0.893 |
| | Happiness | 8 | 0.889 | 0.772–0.914 | 0.873 | 0.762–0.904 |
| | Aggression | 29 | 0.875 | 0.645–0.978 | 0.842 | 0.655–0.914 |
| | The mismatch between effort and reward | 16 | 0.892 | 0.879–0.997 | 0.812 | 0.886–0.912 |

In Table 8, the descriptive statistics of each of the components and then the leading indicators are shown. It is worth mentioning that the number of participants in the Iran questionnaire is 365, and the number of respondents in the Iraq questionnaire is 250. According to the average and minimum obtained from each component, it can be recognized that Iraqi people attach a stronger role to business ethics. The average obtained for Iraqi respondents is 4.086, and for Iranian respondents it is 3.760. In addition, the minimum average for Iraqi interviewees is 2.813, and for Iranian interviewees it is equal to 2.5.

**Table 8.** The Descriptive statistics of hidden research variables.

| Components | Latin Equivalent | Iran | | | | | Iraq | | | | |
|---|---|---|---|---|---|---|---|---|---|---|---|
| | | Observations | Mean | Std. dev. | Min. | Max. | Observations | Mean | Std. dev. | Min. | Max. |
| Business ethics | BET | 365 | 3.760 | 0.476 | 2.500 | 5 | 250 | 4.086 | 0.405 | 2.813 | 5.000 |
| Mental condition | PSC | 365 | 2.814 | 0.574 | 1.138 | 5 | 250 | 3.005 | 0.489 | 1.276 | 5.000 |
| Aggression | AGR | 365 | 2.576 | 0.744 | 1.138 | 5 | 250 | 2.532 | 0.551 | 1.276 | 4.414 |
| The mismatch between effort and reward | MER | 365 | 3.161 | 0.428 | 1.813 | 4.813 | 250 | 3.475 | 0.423 | 2.250 | 4.625 |
| Happiness | WEL | 365 | 3.266 | 0.431 | 1.250 | 4.875 | 250 | 3.532 | 0.416 | 2.250 | 5.000 |

Aggression in two countries can be said to have the same average. However, the discrepancy between effort and reward is greater in Iraq than in Iran, so the average in Iraq is equal to 3.475, and for Iran it is equal to 3.161. Iraqi people are happier than Iranians. The average calculated for Iraq is 3.532, and for Iran it is 3.266. The psychological condition variable, which results from three components of aggression, the dissonance between effort and reward and happiness, is more than in Iraq. The average mental condition for Iraq is 3.005, and for Iran it equals 2.814.

After that, in Table 9, the correlation between the research components of Iran and Iraq has been obtained. The business ethics component affects all indicators of psychological conditions in Iran. In this regard, business ethics has a negative effect on the components of aggression and inconsistency between effort and reward. On the contrary, the business ethics component positively and significantly affects happiness. In general, business ethics has a negative effect on psychological conditions in Iran at the 99% confidence level.

**Table 9.** The correlation matrix of research components for Iranian data.

| | BET | AGR | MER | WEL | PSC |
|---|---|---|---|---|---|
| | | | **Iran** | | |
| BET | 1.000 | | | | |
| AGR | −0.250 *** | 1.000 | | | |
| MER | −0.138 ** | 0.513 *** | 1.000 | | |
| WEL | 0.378 *** | 0.011 | 0.294 *** | 1.000 | |
| PSC | −0.158 *** | 0.248 ** | 0.348 *** | −0.475 *** | 1.000 |
| | | | **Iraq** | | |
| BET | 1.000 | | | | |
| AGR | −0.076 | 1.000 | | | |
| MER | −0.142 ** | 0.428 *** | 1.000 | | |
| WEL | 0.326 *** | 0.304 *** | 0.371 *** | 1.000 | |
| PSC | −0.427 *** | 0.159 ** | 0.182 ** | −0.176 *** | 1.000 |

Notice: Symbols ** and ***, indicate significance levels of 95% and 99%, respectively.

Similar to what was obtained regarding Iran's data, the business ethics component affects all three indicators of psychological conditions in Iraq. Business ethics has a negative effect on the components of aggression and inconsistency between reward and effort. On the contrary, the business ethics component positively and significantly affects happiness. In general, business ethics has a negative effect on psychological conditions in Iraq at the 99% confidence level.

In Figure 1, the output and the effect of the hidden and obvious variables of the Iranian questionnaire are drawn. According to the output of PLS software, the business ethics component affects all three indicators of psychological conditions in Iran. The coefficient of the happiness variable is equal to −0.475, which is significant at the 99% level. On the other hand, the coefficients of inconsistency variables between effort and reward and aggression variables were calculated as 0.348 and 0.248, respectively, which are significant. In general, business ethics has a negative effect on psychological conditions in Iran at the 99% confidence level. The coefficient of this variable is equal to −0.158.

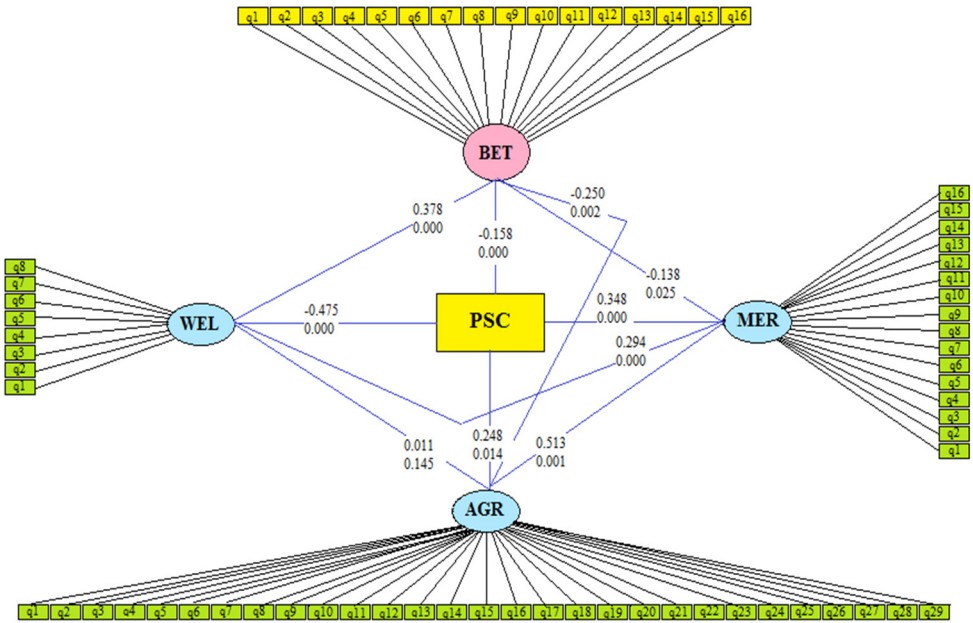

**Figure 1.** The effect of explicit and implicit variables of Iran's data.

In Figure 2, the output and the effect of the hidden and obvious variables of the questionnaire are drawn. According to the output of PLS software, the business ethics component affects two indicators of psychological conditions in Iraq and does not affect aggression. The effectiveness coefficient of the happiness variable is equal to −0.176, which is significant at the 99% level. On the other hand, the coefficients of the inconsistency variables between effort and reward and the aggression variable were calculated as 0.182 and 0.159, respectively, which are significant. In general, business ethics has a negative effect on psychological conditions in Iran at the 99% confidence level. The coefficient of this variable is equal to −0.427.

After that, ordinary least squares regression was used to investigate the relationship between business ethics and psychological conditions. Table 10 shows the results of these regressions for Iranian data. This table contains 4 regressions. In the first model, the dependent variable is the psychological condition (PSC). The variable of business ethics has a negative and significant effect on psychological conditions. With a one percent improvement in the ethics and business variable, mental conditions decrease at the 95 percent confidence level, equivalent to 0.147 percent. The variables of gender and job rank also have a positive and significant effect on psychological conditions at the 99% level. The coefficients of these variables were obtained as 0.179 and 0.471, respectively.

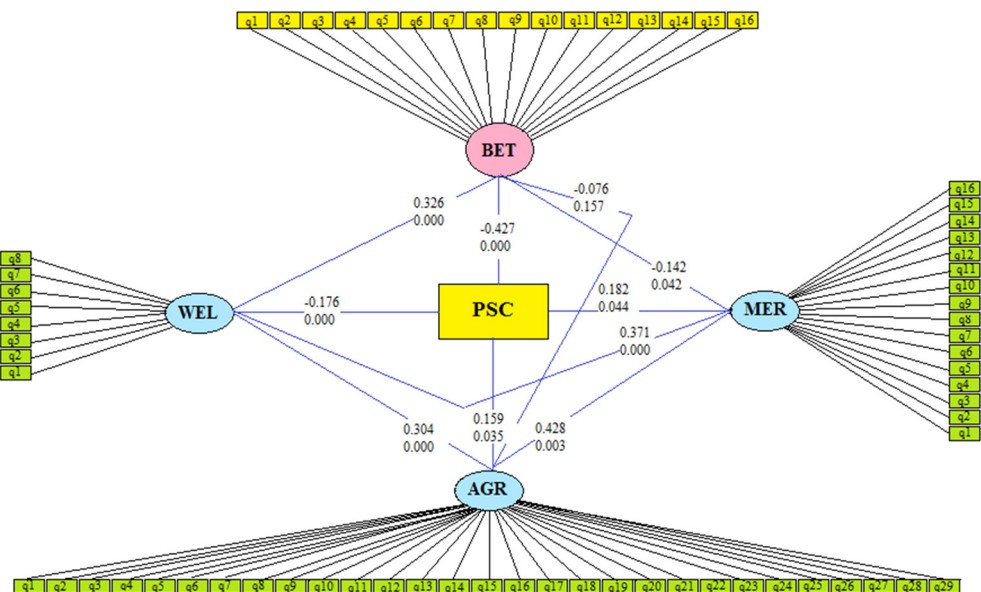

**Figure 2.** The effect of explicit and implicit variables of Iraqi data.

**Table 10.** Regression results of Iranian data.

| Variable | Mode1 1 | Mode1 2 | Mode1 3 | Mode1 4 |
| | PSC | AGR | MER | WEL |
| --- | --- | --- | --- | --- |
| BET | −0.147 ** | −0.372 *** | −0.115 * | 0.335 *** |
| Sex | 0.179 *** | 0.256 ** | 0.215 *** | −0.155 ** |
| Age | 0.022 | 0.038 | −0.017 | 0.027 |
| Education | −0.078 | −0.055 | −0.009 | 0.014 |
| Major | 0.001 | −0.054 | −0.140 ** | 0.029 |
| Experience | −0.031 | −0.056 | 0.0240 | −0.002 |
| Ranking | 0.471 *** | 0.097 *** | 0.029 ** | 0.004 |
| Constant | 2.143 *** | 3.863 *** | 2.766 *** | 1.824 *** |
| R2 Adj | 6.150 | 11.390 | 6.680 | 14.810 |
| Obs | 365 | 365 | 365 | 365 |
| F | 3.850 *** | 6.550 *** | 4.720 *** | 10.040 *** |

Notice: Symbols *, ** and ***, indicate significance levels of 90%, 95% and 99%, respectively.

In the second regression, the dependent variable is aggression. Business conditions have a negative effect on aggression. The coefficient of this variable is equal to −0.372, which is significant at the 99% confidence level. In the third regression, the dependent variable is the mismatch between effort and reward. Business conditions have a negative effect on effort-reward mismatch at the 90% confidence level. In the fourth model, the effect of business conditions on happiness has been measured. According to the obtained results, the variable coefficient of business conditions is equal to 0.335, which is significant at the 99% confidence level.

Considering the data of Iraq, in Table 11 the dependent variable of the first model is the psychological condition (PSC). The variable of business ethics has a negative and significant effect on psychological conditions. Therefore, with a one percent improvement in business ethics in Iraq, psychological conditions will decrease at 99% confidence, equivalent to 0.485 percent. Gender and job experience also increase psychological conditions at the 90% confidence level.

**Table 11.** Regression results of Iraqi data.

| Variable | Mode1 1 | Mode1 2 | Mode1 3 | Mode1 4 |
|---|---|---|---|---|
| | PSC | AGR | MER | WEL |
| BET | −0.485 *** | −0.246 ** | −0.542 *** | 0.343 *** |
| Sex | 0.478 * | 0.047 | 0.042 | 0.030 |
| Age | −0.075 | −0.063 | −0.105 ** | −0.081 * |
| Education | 0.047 | −0.018 | 0.007 | 0.020 |
| Major | −0.047 | −0.090 | −0.034 | −0.027 |
| Experience | 0.005 * | −0.015 | 0.069 ** | 0.039 |
| Ranking | 0.145 | 0.011 | −0.039 ** | −0.036 * |
| Constant | 3.015 *** | 3.236 ** | 2.885 *** | 2.354 *** |
| R2 Adj | 11.140 | 1.830 | 6.180 | 10.610 |
| Obs | 250 | 250 | 250 | 250 |
| F | 6.790 *** | 0.650 | 2.280 ** | 5.220 *** |

Notice: Symbols *, ** and ***, indicate significance levels of 90%, 95% and 99%, respectively.

In the second regression, the dependent variable is aggression. Business conditions have a negative effect on aggression. The coefficient of this variable is equal to −0.246, which is significant at the 95% confidence level. In the third regression, the dependent variable is the mismatch between effort and reward. Business conditions have a negative effect on effort-reward mismatch at the 99% confidence level. In the fourth model, the effect of business conditions on happiness was measured. According to the results, the variable coefficient of business conditions equals 0.343, which is significant at the 99% confidence level.

## 5. Discussion and Conclusions

The failures of the audit profession surrounding the financial scandals of the last few years have damaged the face, position and performance of auditors in society, so trust in this profession has been greatly reduced. In this regard, some researchers have stated that the culture of audit firms has changed in the last few years towards the increasing prioritization of commercial goals (Gendron et al. 2006; Hanlon 1996), though the ideas of officially maintaining the public interest are still evident in implementation. As a result, failure in auditing has been attributed to the lack of professional identity and sufficient prejudice to the code of professional conduct on the part of auditors (Clikeman et al. 2001). The foundation of corporate ethics dates back to the 1980s in the USA. Before the 1980s, the "Corporate Social Responsibility" movement addressed enterprises' social responsibility. Later, the questions of business ethics became discussed with the issue of corporate responsibility. Thus, both corporate and ethical aspects became necessary. On the other side, they also questioned the ethical behavior of companies: "Ethical considerations are often missing from theories of traditional economics. However, thanks to modern economic trends, ethics now plays a vital role in the life of companies, as well as in the field of knowledge management. Unfortunately, to this day, it is still common for some companies to behave unethically in certain practices and even cover them up with untrue statements that are obvious to everyone. Still, everyone pretends to be unaware of them".

In many cases, they proceed in an opposite way to the values revealed (Zsigmond et al. 2022). Therefore, in this research, the relationship between business ethics and happiness, aggressiveness and inconsistency of effort and reward of auditors in the two emerging countries of Iran and Iraq, which have geographical borders close to each other, has been investigated. The results of the hypothesis testing showed a negative and significant relationship between business ethics and auditors' aggressiveness in Iran and Iraq. This aligns with the research results of Gubler et al. (2018), who showed that morality reduces aggression. In such a way, the lack of moral commitment or unethical behavior increases physical aggression (Gubler et al. 2018) and provokes negative emotions, i.e., distress or aggression. Therefore, in order to rely on the results of the companies presenting their reports, auditors may experience psychological anxiety or aggression when faced

with crises and ethical issues, including companies incorrectly reporting and their non-responsiveness to their responsibilities, because the non-compliance with ethics by the companies will cause incorrect and unreliable reports by the auditors and as a result affect their reputation.

Also, the research showed a negative and significant relationship between business ethics and the inconsistency of auditors' efforts and rewards in Iran and Iraq. Since the livelihoods and interests of auditors rely on receiving fees from their employers (Mautz and Sharaf 1961), ethical reasoning by auditors from the environment of the employer's company can influence their thinking about the degree of inconsistency between the effort expended by them and the reward received. Therefore, in the case of a bankruptcy such as Enron, it can be said that this bankruptcy is the result of the participation of audit firms with the leadership of Enron to hide the violations. Hence, according to the analysis of this incident, it can be acknowledged that audit firms try to hide false reports and some immoralities and frauds of companies to receive rewards or fees (Hazgui and Brivot 2020). This paper studied the relationship between business ethics and happiness and found a positive and significant relationship between business ethics and auditors' happiness in Iran and Iraq. This is in line with the research results of Mili et al. (2019), who showed that ethical principles positively affect companies' financial performance. Because desirable ethical behavior affects employees' attitudes and behavioral results such as organizational commitment, deviance, job satisfaction, employee happiness, role performance and creativity, as well as the psychological conditions of other stakeholders of that company, it also affects the psychological conditions of auditors, in such a way that a positive insight regarding the positive results of the company in the case of compliance with ethics causes the auditor to be satisfied with the audit and as a result increases the credibility of the auditor, which will cause the auditor to be happy and vice versa. Negative insight regarding the company's results, in the case of non-compliance with ethics, causes the auditor to be dissatisfied with the audit. As a result, the auditor's credit will be reduced, which will cause the auditor's dissatisfaction. Since the two countries of Iran and Iraq have relatively equal conditions in the economic and political fields, and also because of the proximity of these two countries, the results of the present research in the two countries were the same. According to the results of the research, which showed the positive effect of ethics on reducing aggression and inconsistencies and increasing the happiness of auditors, it is suggested that auditing institutions in terms of improving their position within the profession and increasing the competitive advantage resulting from the improvement of audit quality as a result of growing happiness pay attention to ethical issues and use motivational solutions to improve ethics in their institutions. This study can make a valuable contribution to examining business ethics and the psychological conditions of the auditing profession by evaluating the auditors' moral reasoning abilities and their impact (business ethics) on the auditors' mental conditions in different environmental conditions from the perspective of ethics. These findings will have significant implications for educators in organizations regarding developing and implementing programs and curriculum design centered on business ethics. Professionals of the community of certified accountants should seek to gain people's trust and confidence in this profession. In this way, the accounting community should seriously empower the recruitment, training and preparation of community members of certified accountants to start their businesses with ethical characteristics. Negative characteristics that may affect the knowledge of business ethics of CPA community members may indicate that additional ethical training is needed to resolve ethical dilemmas. By understanding the deficiencies in the ethical business skills of members of the CPA community, the accounting community can use the results as a benchmark to identify those individuals most in need of ethical training.

**Author Contributions:** Conceptualization S.M.M.A.-A. and M.M.; methodology, S.M.M.A.-A.; software, S.M.M.A.-A.; validation, S.M.M.A.-A., M.M.; formal analysis, S.M.M.A.-A.; investigation, S.M.M.A.-A.; resources, S.M.M.A.-A.; data curation, S.M.M.A.-A.; writing—original draft preparation, S.M.M.A.-A.; writing—review and editing, M.M.; visualization, M.M.; supervision, M.M.;

project administration, M.M.; funding acquisition, M.M. All authors have read and agreed to the published version of the manuscript.

**Funding:** This research received no external funding.

**Informed Consent Statement:** Not applicable.

**Data Availability Statement:** Not applicable.

**Conflicts of Interest:** The authors declare no conflict of interest.

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
