# Peer review of "The Effect of Ethics in Business on Happiness, Aggressiveness and Inconsistency of Efforts and Rewards"

_jrfm, doi:10.3390/jrfm16030195_

Round 1

Reviewer 1 Report

The combination of qualitative analysis and quantitative methods of processing its results is positive. However, it would be worthwhile to provide calculating Cochran's sample size for selected countries. 

After all, it is not clear from the phrase 290-291 (Out of 450 questionnaires, 365 were completed by Iranian respondents, and Iraqi respondents completed 250 questionnaires in 2020) what percentage of respondents provided answers by country. Also, the number of respondents given in this phrase does not correspond to the number given in Tables 1, 2 and 3.

The authors noted that these two countries of Iran and Iraq have relatively equal conditions in the economic and political fields, and also because of the proximity of these two countries, the results of the present research in the two countries were the same. However, in the "discussion and conclusions" part, it would be worth adding limitations related to the possibility/impossibility of using research results in countries with different cultural traditions, different legislation, and norms, as well as other traditions of the business environment.

Also, the conclusions should include the prospects of practical application of the obtained results.

Author Response

Dear Reviewer,

Thank you very much for your comments on the paper. The following corrections are made in the current version:

The combination of qualitative analysis and quantitative methods of processing its results is positive. However, it would be worthwhile to provide calculating Cochran's sample size for selected countries. 

With the suggestion of one of the respected reviewers, Cochran's formula was completely removed from the research.

After all, it is not clear from the phrase 290-291 (Out of 450 questionnaires, 365 were completed by Iranian respondents, and Iraqi respondents completed 250 questionnaires in 2020) what percentage of respondents provided answers by country. Also, the number of respondents given in this phrase does not correspond to the number given in Tables 1, 2 and 3.

Thanks to the attention of the honorable reviewer, the exact number of Iranian and Iraqi respondents was determined and corrected. The values of tables 1, 2 and 3 were also modified.

The authors noted that these two countries of Iran and Iraq have relatively equal conditions in the economic and political fields, and also because of the proximity of these two countries, the results of the present research in the two countries were the same. However, in the "discussion and conclusions" part, it would be worth adding limitations related to the possibility/impossibility of using research results in countries with different cultural traditions, different legislation, and norms, as well as other traditions of the business environment. Also, the conclusions should include the prospects of practical application of the obtained results.

Related content as well as practical suggestions, were added to the discussion and conclusion.

Reviewer 2 Report

Synopsis:

The study examines the effect of business ethics on happiness, aggression and inconsistency of effort and reward auditors in Iran and Iraq. The study results indicate a negative and significant relationship between business ethics and aggression, effort-reward mismatch, and a positive and significant relationship between business ethics and happiness.

Comments and suggestions:

However, this study fails to provide detailed information explaining (or developing) the relationships among the proposed constructs. An expert should proofread the paper to remove grammatical mistakes and inconsistencies. I hope my comments and feedback will help improve the merit of this paper.

Introduction

-    The authors need to articulate their research questions/objectives, identify the potential theoretical, background and theoretical motivation or gaps, and explain how your study contributes to the literature. What is the new thing we should know about the business ethics, happiness, aggression, etc. ignored by the existing literature? Authors should straightway start with highlighting these important issues and clearly state the contribution of the study.

-    You can write your introduction in about three paragraphs of 200-250 words each. In the first paragraph define keywords and review the state of the art for your topic, in the second focus on what is missing in the current research, either conceptually or methodologically, and in the third, outline how your article will close that knowledge gap, including justifying the structure of your article to do so.

Literature review

-    The authors need to review the theoretical discussion for the development of all hypotheses. The authors need to explain better these hypotheses. In the current version of the paper, the ideas presented do not lead to a logical understanding of the relationship between business ethics, happiness, aggression, etc. There is a missing of current studies that can support the link between the constructs.

-    Page 4, the authors mentioned “According to the mentioned contents, the second hypothesis of the research is explained as follows”. I am not sure where is the first hypothesis to start from the second, this goes also to the third and fourth.

-    By checking the articles discussed in this study either in the introduction, LR, and methodology all to somehow not more recent for example only one article 2022, two 2021, etc. So, please update the literature used in this study.

-    The methodology appears not to be sound. Detail is needed to highlight, the process to select the sampling and why they focused on the specific places (i.e., Iran and Iraq)?  The measurement items are not clear, and in the research findings there are a lot of tables which make the readers so confused to understand the main findings.

Conclusion and Implications of the study

-    The conclusion part seems like a part of literature review and introduction.

-    The implications are too general and did not offer anything new or interesting. I would like to see the differences and similarities between the findings of this research and those of previous research. How do the findings of this research differ from previous research?

-    The discussions and implications have not been clearly linked back to the hypothesis. Each hypothesis should be clearly discussed again to create a specific implication.

-    The authors should elaborate more on the study implications such as theoretical, practical, and policy implications. 

I wish the authors all the best with their study and hope that my comments and suggestions will be useful in taking their paper to the next level.

 Regards 

Author Response

Dear Reviewer,

Thank you very much for your good comments on the paper; further, the following corrections are made in the current version:

-    The authors need to articulate their research questions/objectives, identify the potential theoretical, background and theoretical motivation or gaps, and explain how your study contributes to the literature. What is the new thing we should know about the business ethics, happiness, aggression, etc. ignored by the existing literature? Authors should straightway start with highlighting these important issues and clearly state the contribution of the study.

Thanks to the opinion of the respected reviewer, all the things that were said were completely corrected and added to the text and highlighted.

-    You can write your introduction in about three paragraphs of 200-250 words each. In the first paragraph define keywords and review the state of the art for your topic, in the second focus on what is missing in the current research, either conceptually or methodologically, and in the third, outline how your article will close that knowledge gap, including justifying the structure of your article to do so.

Thank you for your suggestion. The said comment was done.

Literature review

-    The authors need to review the theoretical discussion for the development of all hypotheses. The authors need to explain better these hypotheses. In the current version of the paper, the ideas presented do not lead to a logical understanding of the relationship between business ethics, happiness, aggression, etc. There is a missing of current studies that can support the link between the constructs.

Relevant and updated content was added to the text.

-    Page 4, the authors mentioned “According to the mentioned contents, the second hypothesis of the research is explained as follows”. I am not sure where is the first hypothesis to start from the second, this goes also to the third and fourth.

A separate explanation was given for each hypothesis.

-    By checking the articles discussed in this study either in the introduction, LR, and methodology all to somehow not more recent for example only one article 2022, two 2021, etc. So, please update the literature used in this study.

References and new research from 2022 and 2023 were added to the research.

-    The methodology appears not to be sound. Detail is needed to highlight, the process to select the sampling and why they focused on the specific places (i.e., Iran and Iraq)?  The measurement items are not clear, and in the research findings there are a lot of tables which make the readers so confused to understand the main findings.

The number of questionnaires and respondents was determined precisely. With the recommendation of one of the respected referees, Cochran's formula was removed from the research. The related tables were merged as much as possible to reduce the number of tables and reduce the confusion of the readers.

Conclusion and Implications of the study

-    The conclusion part seems like a part of literature review and introduction.

The conclusion section was modified and related content was added.

-    The implications are too general and did not offer anything new or interesting. I would like to see the differences and similarities between the findings of this research and those of previous research. How do the findings of this research differ from previous research? The discussions and implications have not been clearly linked back to the hypothesis. Each hypothesis should be clearly discussed again to create a specific implication.  The authors should elaborate more on the study implications such as theoretical, practical, and policy implications. 

Thanks to the opinion of the respected reviewer, all the items presented in the text were added

Reviewer 3 Report

Dear Authors,

find my comments attached. I have some issues:

- you should use more "fresh" studies in Introduction

- the "Methodology" part needs to be some completed (Cochran - I think it should be deleted since, your study does not fill this criteria - the no. of responses is less than 385 in Iraq)

- please add "Limitations" and "Future directions"

My other comments are attached.

All the best!

Author Response

Dear Reviewer,

Thank you very much for your comments on the paper. Further, all comments are incorporated as follows:

There are some formatting issues which have to be corrected:

  1. Please change the color at e-mail addresses to “black”. In addition, the authors should use

their academic emails not “Gmail”. But I leave it on authors.

 The color of Gmail changed to black. And academic email was added for authors.

  1. The “Introduction should be divided into more paragraphs. It is not easy to read this way.

The introduction was divided into more paragraphs

  1. There is something wrong with the hypothesis numbers. First you wrote that “…the second

hypothesis of the research…” and then you label it as “H1”. After that “…the third

hypothesis of the research…” then you label it “H2”. Finally “…the first hypothesis of the

research…” and you refer to it as “H3”

Thanks to the accuracy of the honorable reviewer, the numbers of the assumptions were corrected and highlighted

  1. Tables should not be divided.

The related tables were merged as much as possible to reduce the number of tables and reduce the confusion of the readers.

  1. There is no space above “Table 4.”

The distance created was removed.

Content issues:

  1. In methodology you state that “Out of 450 questionnaires, 365 were completed by Iranian

respondents, and Iraqi respondents completed 250 questionnaires in 2020.” So

365+250=450? I don’t get it.

This part was modified in the research methodology.

  1. In addition, you should add the specific time frame – it is not enough to state that “in 2020”. Which months? It is not the same if you did the research in January-March of 2020 or at the end of the year (because Covid-19 “hit” differently at the beginning).

In fact, the questionnaire of the present research was distributed online from December 27, 2021, and the answers to the questionnaires were collected until the end of March 2022. In general, it can be said that the research was conducted in the first quarter of 2022.

  1. In the case of “Cochran formula” the original source could be mentioned: “Cochran, W.G. (1977), Sampling techniques, John Wiley & Sons, New York, NY.

According to the recommendation of the honorable reviewer, Cochran's formula was removed from the research.

  1. In addition, you describe the formula but you don’t declare that exact number? (It is 385 using Cochran’s calculation). Also, if you want to compare the two countries, then you need at least 385 from each. (I recommend to delete the Cochran part from the study since it does not fulfill its required number).

Thank you very much for the suggestion of the honorable reviewer. Cochran's formula was removed from the research.

  1. The number of mentioned previous studies from 2021-2022 is quite low. Although, you state that: “In recent years, one of the important variables or factors that have attracted the attention of financial and accounting researchers (especially auditors) is ethical decision- making and ethics (Sheehan and Schmidt, 2015) and social responsibility in the audit pro- fession regarding compliance with ethics”. I recommend you the study of Zsigmond et al. (2022) in which they state the following: “Foundation of corporate ethics dates back to the 1980s in the USA. Before the 1980s, the movement called “Corporate Social Responsibility”, dealing with social responsibility of enterprises addressed the issue. Later, the questions of business ethics became discussed with the issue of corporate responsibility, and thus both corporate and ethical aspects became important”. On the other side they also questioned the ethical behavior of companies: “Ethical considerations are often missing from theories of traditional economics. However, thanks to modern economic trends, ethics now plays a key role in the life of companies, as well as in the field of knowledge management. Unfortunately, to this day, it is still common for some companies to behave unethically in certain practices and even cover them up with untrue statements that are obvious to everyone, but everyone pretends to be unaware of them. In many cases, they proceed in a completely opposite way of the values revealed”. You can find more thoughts in their work – I leave it for consideration.

Thank you very much for the guidance of the honorable judge. Thank you for the information you gave me and we used it in ourresearch.

  1. The “Limitations” and “Future directions” are missing from the end of the paper. These are quite important parts!

The mentioned items were added at the end of the conclusion section

Round 2

Reviewer 2 Report

The authors addressed all my previous comments 

Reviewer 3 Report

Dear Authors,

thank you for the opportunity to read your paper! You take my comments into account and the article improved significantly!

There are some minor issues like "AKDAÄžCIK" is with all caps. Also there are blank lines in the "Discussion and Conclusion" chapter. Maybe the editing team will deal with these issues.

All the best!